# Making Every Single Puff Count—Simple and Sensitive E-Cigarette Aerosol Sampling for GCxIMS and GC-MS Analysis

**DOI:** 10.3390/molecules28186574

**Published:** 2023-09-12

**Authors:** Alexander L. R. M. Augustini, Christopher Borg, Stefanie Sielemann, Ursula Telgheder

**Affiliations:** 1Department Hamm 2, Hamm-Lippstadt University of Applied Sciences, Marker Allee 76-78, 59063 Hamm, Germany; alrm@augustini.org (A.L.R.M.A.);; 2Faculty of Chemistry, Instrumental Analytical Chemistry, University of Duisburg-Essen, Universitätsstraße 5, 45141 Essen, Germany

**Keywords:** ion mobility spectrometry, gas chromatography, headspace, sampling, electronic cigarettes, aerosol, vaping

## Abstract

The analysis of the aerosol from tobaccoless electronic cigarettes (e-cigarettes) is an important part of understanding their impact on human health, yet sampling aerosol from e-cigarettes is still considered a challenge. It lacks a standard method for research and quality control and there are a variety of methods. However, few are simple and inexpensive, and none have been suggested for the use with gas chromatography coupled ion mobility spectrometry (GCxIMS). This work presents and evaluates such a setup made from standard lab equipment to quickly collect a quantitative sample from the aerosol of a single puff (5 s totaling 125 mL). The aerosol condensates directly in the cooled headspace (HS) vial, which is analyzed in the HS-GCxIMS or mass spectrometer (HS-GC-MS). The combined use of GC-MS and GCxIMS allows the simple and sensitive identification of unknown substances in complex mixtures and the identification of degradation products in the aerosols. A calibration of 26 flavor compounds (0.2–20 µg/g) was created using single puffs of a spiked, flavorless commercial refill solution and 2-alkanones as internal standards. This sensitive but easily reproducible setup enables a wide range of further investigations, even for labs that were previously unable to afford it.

## 1. Introduction

People using tobaccoless electronic cigarettes (e-cigarettes) have become a common sight in recent years. Most noticeable are the large white clouds which ascend from the users. However, these clouds are formed by heat-induced vaporization of a so-called e-liquid. These e-liquids typically are a mixture of propylene glycol (PG), glycerin (GL), flavor compounds, and often nicotine. Inside the e-cigarette, a coil heats these e-liquids. By inhalation through a mouthpiece, the user creates an airflow along this coil, which allows the formation of the flavored aerosol. This aerosol is then inhaled and allows the user the consumption of the ingredients. This process is known as ’vaping’ [1]. ’Vaping’ is considered less harmful than regular cigarettes, as nicotine is delivered to the user without the combustion products of tobacco. They are often sold as smoking cessation aids [2,3,4,5]. However, the combination of the device, the way it is operated, and the substances it is filled with contributes to the composition of the emissions. The resulting variety demands the analysis of the emissions of e-cigarettes themselves to evaluate the resulting health risk [6].

Through time, the e-cigarettes have been developed from cigarette-reassembling models (first generation) to more adjustable and modular types that offer many options for individualization (third generation). For this work, a third-generation e-cigarette was used. These devices are, due to the modular format with two main parts, very distinguishable. The battery segment, containing the power storage and control electronics, is the largest part of the e-cigarette and is used to hold it. The head or the ’atomizer’ consists of the mouthpiece, the reservoir for the e-liquid, and the coil. The e-liquid is usually stored in a glass chamber around the coil and is connected to it using a wick, which feeds the coil. Due to its design, it is sometimes called a clearomizer [7,8].

Due to the low resistance of the coil that is used in the e-cigarette (<1 Ω), this type of e-cigarette is known as a sub-ohmic type and allows the use of more power. This is known to aerosolize larger amounts of e-liquid and has been described to increase the amount of degradation products in the aersol [9,10].

Current analyses have been focused on two main parts: the refill solution (e-liquid) and the consumed aerosol. The e-liquid can be analyzed as easily as any other matrix-heavy liquid. These samples are often diluted and injected directly [11,12]. In the case of gas chromatography (GC) and the analysis of volatile organic compounds (VOC), the use of headspace (HS) sampling [13,14] or solid phase microextraction (SPME) [15] is also quite common.

In contrast to sampling from the liquid phase, the gaseous phase introduces other challenges, e.g., aerosol handling and the low analyte content in large volumes. Sampling smoke from cigarettes is well established with the help of smoking machines collecting samples with glass filter pads and sometimes additional impingers. However, while these methods work well with regular cigarettes, they can be less efficient when used for e-cigarettes [6,16], as the substances of interest and the matrix in e-cigarettes differ [17]. Therefore, other sampling aids have been tested to remediate this discrepancy. These include cartridges coated with derivatization agents [18], silica gel traps [19], polymer sorbent tubes [20,21,22,23], SPME fibers in a static gaseous sample [24], solvent-filled impingers [25,26,27] or a combination of these [28,29]. As with regular smoking machines, the necessary suction is produced by a simple vacuum pump or a large-volume syringe pump.

The collected samples on sorbent tubes are then either directly analyzed using thermal desorption units (TD) [22,23] or comparable to the coated cartridges eluted with solvent and then analyzed [18,20,21]. Especially the suitability of the coated cartridges is being discussed, as the smallest substances tend to be very reactive [30].

The use of solvent-filled impingers results in a dilution of the collected analytes in a large volume of a solvent. This dilution requires collecting larger amounts of aerosol [25,28,31]. Unfortunately, the trapped substances can be unstable, leading to underestimations when using this method [31]. Specialized sorbent tubes also have high analyte capacities and are used with an intense puffing regime as well [7,21].

Modern e-cigarettes limit puff length to reduce the risk of damage to the device by overheating. Using repeated puffs for a single sample up to draining the entire e-liquid stored in the reservoir is a way to increase the amount of sample in a vapor trap while using realistic puff sizes and flow rates [2,19,32]. However, this procedure differs from a consumer’s usage and increases the coil’s risk of running dry. When the e-cigarette is operated without sufficient e-liquid feeding the coil, a ‘dry puff’ can occur. The aerosol formation from the e-liquid cools the coil, so a coil without enough e-liquid overheats. An overheating coil leads to combustion products forming from the remaining e-liquid and the wicking material. Consumers describe the inhalation of a dry puff, a so-called ‘dry-hit’, as unpleasant. Therefore, it is avoided under normal conditions. When sampling automatically, this has to be monitored, as a dry puff drastically changes a sample’s composition [33,34]. Additionally, it has been found that the e-liquid composition can change during consumption [35], thus increasing the relevance of single-puff analysis.

For sensitive flavor analytics, GC-coupled mass spectrometry (GC-MS) is a common tool [36]. Thus, it is well established for the analysis of e-cigarettes [11,23,37,38,39]. However, sensitivity and resolution can be an issue when scanning a sample. Flavor compounds tend to be structurally similar, thus fragmenting similarly when ionized using electron impact ionization (EI). This makes it more difficult to identify coeluting substances and decreases the separation power for individual substances. MS with a triple quadrupole or a higher resolution can resolve this problem at the cost of the more elaborate setup [40].

An ion mobility spectrometer (IMS) separates ionized analytes using their mobility in an electric field and a gaseous atmosphere at ambient pressure. This separation by collision cross-section allows the separation of isomers [41]. IMS is well known for its sensitivity for small, polar compounds [42] due to the ionization by radioactive atmospheric pressure chemical ionization (R-APCI). In this case, tritium is used as a source of beta radiation to indirectly create ionized analyte–water clusters [41]. Since the formation of these clusters depends on the substances present in the ionization area of the IMS, the spectrum of an IMS measurement contains different signals. These signals are primarily the reactant ions which are just the ionized water clusters and the various ionized analyte–water clusters which can contain one or more of the same or different analyte molecules, resulting in the formation of monomers and (mixed) dimers [43,44,45].

The IMS technology has been used for decades to detect dangerous chemicals, including explosives, drugs, and chemical warfare agents [46,47,48,49,50,51]. It is most commonly known for its use as a fast and on-site detection system for explosives at airports. Especially when coupled to a GC, the field of application is constantly growing [52,53,54]. The detection of dangerous chemicals is still an important application in the monitoring of environments [55]. However, more lab-based applications are also on the rise [56]. Many of these applications concern volatile components of foodstuff [57,58,59,60]. Therefore, the application of this technology for the analysis of flavor compounds in e-cigarettes is self-evident. The use of headspace samples has been shown to work very well for this kind of analysis [61,62,63].

This work aimed to develop a quick and straightforward sampling setup to collect the emission from e-cigarettes for analysis. The e-cigarette aerosol was successfully condensed directly in an HS vial during a preliminary feasibility study. Based on these results, this work’s main goal was refined to build a simple sampling setup through condensation and assess its validity and reproducibility. The sampling system was built using readily available lab materials to allow for a simple recreation in other labs. The working prototype was tested using a versatile commercial e-cigarette with e-liquids of known composition. This allowed the optimization of the setup to produce the most reliable results.

Various parameters were tested for their influence on the sampling process and the results, including condensation aids, puff length, flow rate, and vaping temperature. The analysis was conducted on a double-flow-line-GC with a static HS-autosampler, detecting the analytes using a drift-tube IMS (DT-IMS) or a MS, as described in a previous work [61]. This allowed for the combination of the sensitive and powerful separation of the GCxIMS with the additional identification power of the GC-MS [62].

## 2. Results and Discussion

Using our prototype of an efficient sample condensation system, aerosol samples were collected from a commercially available e-cigarette. This allowed a simple analysis of the chemical composition of the puffs, focusing on identifying and quantifying individual substances.

The aerosol was created from ready-to-use and self-made e-liquids using a third-generation e-cigarette and a vacuum pump, creating the necessary suction. While flowing through an HS vial submerged in a salt-ice bath, the aerosol condensed and was collected. There is a detailed description, including a scheme, of the sampling setup and process in Section 3.2 and Section 3.3, with a photo in Figure A1 in the Appendix A.

The vials were then used for the analysis by GCxIMS and GC-MS. The sampling process was optimized to achieve the best possible reproducibility and sensitivity. The effects of condensation aids, the composition of the cooling bath, the flow rate, and the puff length were evaluated. As samples were single-use only and manual sampling is quite time-consuming, the collected samples during the optimization process were analyzed on the GC-MS, as this is the established benchmark technology for the analysis of flavor compounds.

The adjustments of the sample collection are described in the following paragraphs, whereas the analysis by HS-GCxIMS and HS-GC-MS was based on the previously optimized and published method [61,62,63] and is described in Section 3.4 in detail.

Figure 1 shows the resulting GCxIMS-plot of a collected aerosol sample from a store-bought blueberry-flavored e-liquid after the optimization process. It displays the two-dimensional separation based on the retention time from the GC and the drift time in the DT-IMS. Next to it, the corresponding GC-MS total ion chromatogram (TIC) is displayed. The TIC was fitted to the retention time scaling of the GCxIMS using the previously published retention index-based correlation method [62]. This method uses the 2-alkanones as retention time markers to normalize the differing retention times of the GCxIMS and GC-MS so that they can be compared, and the chromatograms can be converted to use the same timescale. However, only the time range covered by these markers (2-butanone to 2-decanone) was available for correlation. Substances were tentatively identified using the correlated mass spectra. When available, reference substances were used to confirm the identification, comparing the retention and drift times from the GCxIMS. The named substances are listed in Table 1 with the different retention times and the reduced mobility as calculated for the IMS [64].

In cases where there was a coelution of the internal standard and an analyte, the results were compared with a measurement without the internal standard present. This allowed the separation of the internal standard and the coeluting substance. This was the case for 2-nonanone, which coelutes in the blueberry-flavored e-liquid with pentyl butanoate in the GCxIMS.

The drift and retention times of the substances were determined to be very stable (±4 µs and ±1 s, respectively) over a period of 60 days, thus allowing for easy comparison with reference measurements. In stable conditions, these parameters should not vary indefinitely. However, like any calibration, this stability has to be checked for in regular intervals.

### 2.1. Cooling Methods

In initial tests, the cooling of the vial was investigated. As expected, when looking at sample collection at room temperature, in an ice bath or a salt ice bath, the colder temperature improves the condensation of the volatile analytes. However, when cooling with dry ice, the intense cold increases the condensation outside the headspace vial. As a result, the e-liquid condensed in the pipette connecting the e-cigarette with the headspace vial, with droplets blocking the pipette tip. This blockage resulted in a significant loss of recovery and reproducibility.

The results of the GC-MS measurements are shown in Figure 2 for five repetitions each. The logarithmic scale was chosen to cover the entire observed intensity range for the different substances. The higher signal intensities and lower standard deviations (as represented by the error bars) favored using the salt–ice bath as the cooling method, which was chosen for further analyses with this setup.

Using a more elaborate setup with an insulated or heated transfer line into the vial might reduce the condensation outside the vial, thus improving the sampling conditions. However, such a setup would have exceeded this work’s scope and limited the ability of others to recreate this sampling method.

### 2.2. Condensation Aids

A logical step to compensate for the low concentration of the analytes in the vapor is using means to improve their condensation in the headspace vial. Therefore, one of the modifications investigated during the method development was using a condensation aid in the headspace vial. Two fillings were chosen to improve the condensation: First, loose glass wool was added into the vial to increase the surface on which the aerosol could condensate without changing the underlying chemistry. This procedure was inspired by multiple publications using glass wool filter pads in vaping machines [6,16,25,31,32].

As an alternative, activated charcoal pellets were filled into the vial before sampling. Both filled-type vials were tested and compared to a vial without condensation aids. The results from the GC-MS measurements are presented in Figure 3. Most apparent is the lack of flavor compounds in the sample containing activated charcoal. Only the intense GL peak proves the presence of aerosol in the sample. The assumption was that the maximum incubation temperature of the headspace vial (150 °C) and the static atmosphere did not suffice to desorb the flavor compounds from the activated charcoal, thus rendering this type of trap useless for this autosampler. Other adsorption aids like Tenax or chemosorbs have been described with similar requirements [65]. Therefore, this type of enrichment for the VOCs was rejected.

In contrast, the glass wool shows promise, as many flavor compounds were detected in these samples. However, closer examination shows no improved recovery compared to the empty vial. This can be due to the headspace vial’s changed flow path. As the gas enters and exits the headspace vial at the top, the glass wool reduces the volume the aerosol can fill quickly.

Thus, glass wool may be a possible condensation aid when the flow path is directed through it, like in the previously mentioned publications, using a glass filter pad. This would also improve the retention of particles that may interact with the analytes. However, the current setup does not profit from its use. The gas flow needs to be channeled through a highly reproducible glass wool filling to use its full potential, but this modification was not possible for this easy-to-use setup. Therefore, in further experiments, the headspace vials were used without any fillings.

### 2.3. Sampling Procedure

Another essential part is the sampling procedure itself. It turned out that it is crucial how the sampling is stopped and the vial is handled. Optimization aims to minimize the loss of collected sample material while achieving reproducible results. In principle, using HS vials for the sampling process allows direct analysis without any necessary changes. However, the septum of the vial is perforated during the sampling process and leaks when the pressure inside the vial is increased during the incubation. Therefore, to reduce the loss of volatile compounds, the vial cap was immediately exchanged for a new one after the sampling process was finished. The vial was kept in the salt–ice bath for this.

In initial investigations, another primary influence was identified to be the way to stop the sampling process. A vacuum pump creates the necessary airflow to activate the e-cigarette and draw the sample. However, it is slow to react, and the large volume in the system slows the response on the sampling process even more.

The production of aerosol by the e-cigarette can be controlled accurately by using its power button. However, when the aerosol production is stopped while there is still a low pressure at the outlet of the vial, the airflow through the vial continues and removes part of the sample from the vial again. Thus, to precisely control the sampling and to achieve reproducible results, it is necessary to cut off the vacuum as quickly as possible.

A valve was initially included in the setup to control the airflow precisely. However, there was no ideal position of the valve in the flowline. When the valve was positioned directly at the outlet of the vial, it tended to clog after only a few measurements. When the vial was placed after the washing flasks, the cut-off effect was reduced; whenever the valve was closed, the low pressure inside the washing flasks remained and was equalized by the gas flow through the vial, thus still removing vapor before all the analytes condensed in there. In the end, the cut-off valve was removed, and the sampling was stopped by pulling the out-flow cannula from the headspace vial at the same time the power button of the e-cigarette was released. The final setup is described and shown in Section 3.2.

### 2.4. Puff Length and Flow Rate

A general factor in e-cigarette sampling is puff duration and flow rate. Unfortunately, there are no universally accepted parameters for a puff from an e-cigarette. In the literature, many different lengths and flow rates have been described. A short review provided flow rates between 0.5 and 9.9 L/min [16,21]. However, most publications use flow rates of about 1–2 L/min [18,25,32]. Puff lengths span 2–4 s in most applications [18,21,25,28,32,66]. Generally, regular cigarettes are smoked faster with shorter puff lengths, whereas e-cigarettes are puffed longer, with a slower inhalation [67,68].

During the testing for puff length, it was noticed that long puffs, for example, 10 s, are not advisable, as they rapidly increase the consumption of e-liquid, risking a dry puff. They are also unusual for consumers. When drawing very short puffs, the reproducibility of the sampling drops, as the slight variations due to the manual handling have a greater effect on the sample. Sampling times were reproducible with ±0.2 s in this work. Thus, an average puff length of 5 s was chosen to be practical.

For the evaluation of the effect of the flow rate, samples were taken from an aerosolized, lab-made e-liquid at 0.5–4 L/min. The signal intensities and their standard deviations from three repetitions were compared. The results are plotted for six selected substances in Figure 4. Two different trends are noticeable. For very volatile substances (e.g., ethyl acetate and methylbutanal), the signal intensity varies very little for different flow rates and the standard deviation is at its lowest at 3 L/min. The less volatile substances (e.g., methyl hexanoate, octen-3-one, octanal, and menthol) show increasing signal intensities and standard deviations for higher flow rates.

Substances with a lower volatility and vapor pressure condense quickly in the cold HS vial. Thus, at higher flow rates, when more aerosol passes through the vial, more mass is recovered in the vials. Highly volatile substances with high vapor pressure tend to evaporate quicker at higher gas flows. Since the trapped amount of more volatile substances stays nearly constant at varying flow rates, the major condensation of these more volatile substances should happen after the flow is cut off. Thus, only the amount of these substances, which is in the aerosol that fits into the vial (20 mL), is trapped. Therefore, there is no accumulation of these substances during the sampling time. Nonetheless, high gas flows increase the total amount of gas passing through the vial, thus allowing for a better purge of the initially contained air.

As a compromise, we decided on a 1.5 L/min 5 s puff, resulting in a 125 mL total flow volume for the final analysis. The resulting procedure enabled reproducible measurements and balanced signal intensities for the low and high-volatile substances. However, its flexibility makes other flow rates and puff lengths easily realizable.

### 2.5. Reproducibility of Measurements and the Usage of Internal Standards

In a previous publication [63], 2-alkanones were successfully used as internal standards with the HS-GCxIMS measurements of e-liquids to compensate for the deviations of repeated measurements. This insight was also transferred to the capture and analysis of the aerosolized e-liquids. Using the GC-MS measurements of seven individually trapped puffs (5 s at 1.5 L/min) of a lab-made e-liquid, the reproducibility of the sampling method was evaluated. To choose the best internal standard, each analyte was compared with its two closest eluting 2-alkanones. The ratio of intensities of the analyte and the internal standards were taken, and the best fit was chosen in regard to the previous publication [63]. In Figure 5, the coefficient of variance of the aforementioned seven measurements is plotted for the analyzed substances, comparing the compensated and non-compensated results.

For most substances, using the internal standard reduces the standard deviations of repeated measurements by up to 10%. Later eluting substances, like the terpenes linalool and menthol, see a reduction of over 15%. Only for two substances (diacetyl and isobutanol) the deviations increase when the internal standard (2-butanone) is used. For these two substances, significant impurity is detected by GC-MS. Therefore, they have to be quantified using a different mass peak instead of the base peak (see Table A1 in the Appendix A for the list of the quantifier and qualifier signals). Diacetyl coelutes with 2-butanone, so using a different internal standard can improve this situation. Unfortunately, isobutanol coelutes with a suspected degradation product of PG. Thus, a different separation is necessary to improve this.

### 2.6. Quantitative Analysis

A lab-prepared e-liquid with different concentrations of flavor compounds (approx. 0.2, 1, 5, 10, and 20 µg/g) was used to evaluate the quantitative analysis by applying this sampling method. The internal standard mixture was added for a final content of 10 µg/g for each substance. Six aerosol samples were taken for each concentration level with the e-cigarette at 100 W and 100 °C using a 1.5 mL/min flow. They were collected for 5 s at −20 °C in individual HS vials. These were then analyzed with the GC-MS or GCxIMS.

As discussed in a previous publication [63], quantitation using GCxIMS with r-APCI presents a challenge due to the limited linear range. Nonetheless, the published method linearizes the otherwise non-linear calibration. Thus, this method was applied successfully for 26 substances in the e-cigarette aerosol. Internal standards were used to increase the precision, as described in the previous section.

As seen in Figure 6, a linear relationship between the signal intensity of each detector and the concentration of a flavor compound in the e-liquid is possible. The results are summarized for both analyzing systems in Table 2, showing the root mean squared error (RMSE), the relative error (RE), and the coefficient of determination (r^2^) for the linear regression.

Generally, the direct comparison shows an advantage for the calibration by GC-MS. The linear fit is overall better, as shown by the lower RMSE and RE, as well as the higher r^2^. This was expected, as the dynamic range of the IMS poses a challenge and the signal had to be converted to use the linear regression.

A few substances, like ethyl acetate, isobutanol, ethyl 2-methyl butanoate, 1-octanol, and D/L-menthol, show a better linear fit for the GCxIMS than the GC-MS. However, differences are all below 4% RE, thus well within the expected error range, as shown by Figure 5, and are thus negligible.

Figure 1 shows the primary challenge when analyzing an e-liquid. The most dominant signal in the GCxIMS plot and the GC-MS TIC represents PG, one of the major matrix compounds. The other main component of the matrix, GL, is only visible in the GC-MS plot, as the IMS does not detect it.

The PG signal was previously shown to hamper quantitation in GCxIMS [62], so this very dominant substance is also a concern here. Even though Figure 5 shows that a reproducible signal is possible for substances affected by the coelution (e.g., isoamyl alcohol and 2,3-hexanedione) at 10 µg/g, the matrix signal suppresses the analyte signal considerably. This suppression inhibits the usage of the lowest two concentrations for the calibration. Thus, the results for isoamyl alcohol, methyl isobutyl ketone, and 2,3-hexanedione from the IMS were excluded from Table 2.

In GC-MS, substances coeluting with the matrix (e.g., isoamyl alcohol and methyl isobutyl ketone) are not visible in the TIC, as the larger matrix signal covers their peaks. In this case, these substances can be found by looking for the specific *m*/*z*-signals. Unknown substances are hard to detect this way and cannot be adequately identified. In the GCxIMS-plot, these substances are visible next to the PG signal. However, the ionization of these analytes is less suppressed by the competing ionization of the matrix molecules when using the electron ionization of the MS. Therefore, in contrast to the GCxIMS, these substances were available for quantitation, as shown in Table 2.

Additionally, due to its higher response at low concentrations, the PG signal has a considerably longer tailing in the GCxIMS plot than in the GC-MS TIC, as shown in Figure 1. The tailing impacts the quantitation for later eluting substances as well. Using comparably affected internal standards improves the fluctuation of the signal intensities and allows for improved quantitation, as shown in Table 2.

Water can be used to suppress the polar matrix compounds, as shown in a previous publication [63]. Unfortunately, adding water reduces the signal intensities of other compounds as well. Thus, this is a hindrance when looking for trace substances, e.g., reaction products from the degradation in the e-cigarette, which was an important objective of this work.

### 2.7. Reactivity and Degradation

Third-generation e-cigarettes offer consumers more control over the workings of their devices. One essential part is regulating the power supplied to the coil, which influences the coil’s temperature and substantially impacts aerosol creation.

The e-cigarette used in this research also allowed a preset coil’s temperature to be set directly. This mode was used to investigate the reactivity and possible degradation of e-liquid components during vaping. The flavorless e-liquid was used with and without added flavor compounds at three different temperature settings (100, 200, and 300 °C), and the samples were analyzed using GC-MS and GCxIMS. The actual temperature of the e-liquid on the wick is additionally influenced by the evaporation of the matrix components GL, PG, and water [69]. Unfortunately the closed setup did not allow the use of a thermometer to measure the temperature of the e-liquid or the wick while sampling.

Results are presented in Figure 7, which shows excerpts of GCxIMS plots for three flavored samples taken at the mentioned temperatures compared to the analysis of the e-liquid itself. For reference, the GC-MS’s TIC is shown for the 300 °C sample. The effects are most visible for the highly volatile compounds in Figure 7.

A considerable number of additional compounds was detected in the aerosol compared to the e-liquid. When the e-cigarette was run at higher temperatures, the number and intensity of these additional substances increased. As previously described, when the aerosol is created under more intense settings, highly volatile degradation products can be created, including acetoin, acetol, and methacrolein [23,70]. These substances were also tentatively identified here and are shown as no. 2, 6, and 9 in Figure 7.

It is especially noticeable that only very low concentrations of acetol and neither acetoin nor methacrolein were found when a flavorless e-liquid was used. Acetoin and methacrolein were only detected when e-liquids containing additional flavor compounds were vaped. The concentration of acetol increased in these samples as well. This finding leads to the conclusion that flavor compounds drive the chemical reactions observed during the vaping process that create these new substances. Klager et al. have suggested this as well [18].

In addition to the newly formed highly volatile substances shown in Figure 7, which likely are products of degradation processes, unknown signals representing larger molecules have also been detected in these measurements. Since no reference substances were available and the focus was on sampling rather than the elucidation of the degradation process, identifications are rudimentary, awaiting verification by other means.

However, two substances stood out as they were tentatively identified in multiple measurements with sufficient probability: 1,2-Propanediol acetate (eluting at 9.29 min) was found in the aerosol as well as the unvaped e-liquids. Thus, the necessary transesterification probably occured during storage.

The observation of (E/Z)-2-hexenal propylene glycol acetal (eluting at 17.26 and 17.47 min) indicates a condensation between the excess propylene glycol and the (E)-2-hexenal from the flavor mixture. Since the corresponding peaks were not found in the reference measurements of the non-aerosolized e-liquids, this reaction appears to be accelerated by the vaping process. However, acetal formation during storage was observed by Erythropel et al. [71]. This should therefore be taken into account when validating these results. Further considerations can only be the goal of future work, but this initial feasibility study already shows the potential of this sampling and analysis technique.

## 3. Materials and Methods

### 3.1. Samples

A flavorless e-liquid (55% PG, 35% GL, 10% water, and 3 mg/mL nicotine; VapeBase GmbH, Essen) was used as the base, which was spiked with a set of selected flavor compounds (see Table A1 in the Appendix A). An internal standard consisting of 2-alkanones (2-butanone to 2-decanone) was added before the e-liquid was used in the e-cigarette. If not otherwise specified, the analytes and internal standards were added at a level of about 10 µg/g.

### 3.2. Aerosol Production and Sampling Setup

The aerosol was produced by a tank-style rechargeable third-generation e-cigarette (Valyrian II, Uwell, Guangdong, China). It consists of a battery pack with the control electronics and a detachable e-liquid reservoir, in which the atomizer with a single coil is embedded (UWELL Valyrian II UN2 Single Meshed Coil, 0.27 Ω–0.32 Ω). The coil has a resistance < 1 Ω; thus, this type of e-cigarette is known as a sub-ohmic-type. The atomizer and the coil are consumables. When regularly used, they have a lifespan of one to four weeks. If used less, their lifespan can exceed eight weeks. Thus, a single coil was used for several experiments during this research. Since the atomizer, including the coil, can be easily exchanged and cleaned, they were re-used with different e-liquids. Before re-use, the coils were washed four times with isopropanol. The sealings were cleaned separately. Everything was dried afterwards. The e-cigarette was powered by three 18650-type batteries (Konion VTC5A, SONY, Tokyo, Japan), which were fully charged before use.

For operation, the e-cigarette offers three power modes: The variable wattage mode allows only the wattage (50 W–300 W) to be adjusted. The temperature control mode enables the choice of a wattage of 10 W–120 W and a temperature (100 °C–315 °C). Further, the e-cigarette offers a “temperature coefficient of resistance” mode with opportunities for self-wound coils. Finally, the device measured each puff’s length as running time in seconds.

To sample from the e-cigarette, it was directly connected to the sampling setup. The conventional mouthpiece was replaced with a customized adapter made from polytetrafluoroethylene (PTFE). A bent glass pipette was then inserted through the septum (PTFE/butyl rubber) of a 20 mL headspace vial (Machery-Nagel, Düren, Germany) and its wider end was inserted into the PTFE adapter. This setup created a flow path from the e-cigarette into the vial in which the sample was collected. Next, a cannula was inserted into the septum of the vial and connected via Perfluoroalkoxy alkane (PFA) tubing, and two washing flasks to a vacuum pump. The first washing flask was filled with water as a trap for excess aerosol, with the second washing flask used to protect against spilled water. These flasks protected the flowmeter, valve and vacuum pump from contamination.

The suction on the e-cigarette was generated using a lab vacuum pump (PC 3002 Vario, Vacuubrand, Wertheim, Germany) connected with a rotameter-type flowmeter (0.3 L/min–4.6 L/min; Uniflux, VAF Fluid Technik, Lichtenau, Germany) regulating the air flow rate using a simple valve. The suction flow on the e-cigarette was checked using a digital flowmeter (Ellutia 7000, Ellutia, Kassel, Germany). The e-cigarette was fixed with a clamp onto a support stand and could be easily adjusted in orientation to mimic the regular positions during the vaping process. During the operation of the e-cigarette, it was positioned in a near-horizontal orientation. In between uses, it was in an upright position to ensure the wetting of the wicking material, feeding the coil with the liquid. This procedure prevents coil and wicking material from overheating due to insufficiently available e-liquid. During the sampling process, the vial was cooled to temperatures between −15 °C and −21 °C by submerging it in a bath of ice and NaCl. Figure 8 shows a scheme of the experimental setup. A photo of the setup is included in the Appendix A (see Figure A1).

### 3.3. The Sampling Process

For sampling, the e-cigarette’s reservoir was filled with e-liquid samples. The cooled vial was connected to the e-cigarette and the vacuum pump. To start the process, the power button of the e-cigarette was pressed. A continuous flow of aerosolized sample streamed for a fixed time through the vial. The aerosol condensed on the cold walls of the vial. At the end of the sampling period, the vacuum pump was disconnected from the vial by pulling the cannula out of the septum. At the same time, the power button of the e-cigarette was released. Immediately, the pipette was removed from the vial. To ensure that the trapped aerosol condensed completely, the vial was cooled for a few more minutes until no more vapor was visible in the vial. To finally prepare the vial for the headspace analysis, the lid was replaced with a softer, unperforated PTFE/Polysiloxane septum (CS—Chromatographie Service, Langerwehe, Germany). Otherwise, substances could escape through the perforated septum during the incubation for the headspace analysis. The change of the lids was conducted quickly with the continuously cooled vial.

### 3.4. Analysis

After sample preparation, the vials were shaken for 30 min at 120 °C in a heated autosampler for static headspace analysis (AOC 5000 plus, Shimadzu, Kyoto, Japan). From the headspace, 2 mL were automatically injected into one of the two injection ports (each at 150 °C; split: 1:10) of a GC-2010 Plus gas chromatograph (Shimadzu) using two separate flowlines with similar columns (HP-5 MS UI 30 m × 0.25 mm × 0.5 µm, Agilent, Santa Clara, CA, USA), running a constant flowrate of 35 cm/s helium and a custom temperature ramp (held at 50 °C for 4 min; with 5 °C/min to 150 °C; with 10 °C/min to 200 °C, held up to a total measurement time of 35 min).

Detection was performed on either the manufacturer-improved ion mobility spectrometer with shorter drift tube and improved ionization area (G.A.S., Dortmund, Germany; drift tube: 15.2 × 53 mm; temperature: 80 °C; ^3^H-source; 150 mL/min nitrogen as drift gas; field strength: 500 V/cm) or the mass spectrometer QP-2010 Ultra (Shimadzu; EI source at 200 °C with 70 eV; scanning 32–300 *m*/*z* at 3.3 Hz).

### 3.5. Data Processing

GCxIMS measurements were processed with the LAV software suite (Version 2.2.1, G.A.S. mbH) using a Savitzky–Golay filter. The results were displayed as a heatmap showing the relative drift time depending on the position of the RIP (reactant ion peak) on the x-axis, the retention time on the y-axis, and the signal intensity as the color scale. GC-MS measurements were processed with GCMSsolutions (Version 4.30, Shimadzu). Substances were tentatively identified using the fragmentation patterns provided by the GC-MS, the retention indices, and the NIST database (National institute of standards and technology, Gaithersburg, MD, USA, 2014).

Identified substances with available reference standards for an external calibration were quantified using an appropriate *m*/*z*-signal, the quantifier ion. These were selected to be the largest base-separated single *m*/*z*-peak for the respective substance. Additionally, the presence of other fragment signals, known as qualifier ions, was checked to reduce the possibility of mix-ups during the quantitation.

Excel 2019 (Microsoft, Redmond, WA, USA) was used to organize the data and perform simple calculations. Plots were created with OriginPro 2022 (OriginLab, Northampton, MA, USA).

### 3.6. Calculations

Retention times of the GC-MS and GCxIMS measurements were correlated using retention indices based on 2-alkanones as retention time markers using Equation (Equation 1). This formula was originally published by van den Dool and Dec. Kratz in 1963 [72]. The procedure for this application was validated and is described in depth in a previous work [62].

The retention index *I* is calculated using retention times of the sample (*X*) and the earlier and later eluting reference molecules (*M*) with the number of carbon atoms (${}_{n}$) and (${}_{n+i}$), respectively. When needed and using a different formula by van den Dool and Dec. Kratz, this 2-alkanone-based retention index can then be converted into the alkane-based index [62,72].
(1)$$\begin{array}{c} I=100n_{i}\frac{X-M_{n}}{M_{n+1}-M_{n}}+100n. \end{array}$$

GCxIMS and R-APCI allow for only a small range in which the signal intensity linearly increases with a substance’s concentration. As described in Augustini et al. 2022 [63], it is possible to either use a non-linear regression (e.g., Boltzmann-type) or to normalize the signal intensities based on the available reactant ions [63]. The latter option (see Equation (Equation 2)) results in a near-linear relation. This allows for the use of a linear regression and the far more straightforward calculation of the concentration in an unknown sample, which most people are accustomed to.

The ionization of the analyte molecules depends on the available reactant ions (ionized water clusters). Since the amount of available reactant ions decreases with every analyte molecule ionized, this change must be considered.

The calculation is derived from the law of mass action for the ionization reaction. The reaction can be simplified to the point where the concentration of the analyte (A) is linearly dependent on the fraction of the ionized analytes (Sum of the Analyte Ion Peaks $AIP_{S}$) over the remaining reactant ions, which can be calculated as the total available reactant ions (Reactant Ion Peak at the start of the measurement: $RIP_{0}$) minus the ionized analytes ($AIP_{S}$) [63].
(2)$$\begin{array}{c} m\cdot \left[A\right]\approx \frac{AIP_{S}}{RIP_{0}-AIP_{S}}. \end{array}$$

The goodness of fit for the calibration was compared by calculating the concentrations of the analytes using the regression for the results of GC-MS and the linearized results of GCxIMS. The root mean squared error (RMSE, see Equation (Equation 3)), the relative error (RE, see Equation (Equation 4)) and the coefficient of determination (r^2^) were used as parameters for evaluation [73,74].
(3)$$RMSE=\sqrt{\frac{\sum_{i=1}^{n}{\left(c_{i}-c_{i}^{calc}\right)}^{2}}{n}},$$
(4)$$RE=\sqrt{\frac{\sum_{i=1}^{n}{\left(c_{i}-c_{i}^{calc}\right)}^{2}}{\sum_{i=1}^{n}{\left(c_{i}\right)}^{2}}}.$$

## 4. Conclusions

This work presents a simple and fast setup to sample a single puff’s aerosol created by an e-cigarette. Especially the direct collection in HS vials instead of the more common TD tubes makes this setup more affordable, particularly when looking for very high sample numbers or sampling rates. Additionally, the use of condensation aids was tested but rejected due to the loss of reproducibility and sensitivity.

The results from samples with different concentrations showed a direct relationship between the sample’s concentration and the detector’s signal for 0.2–20 µg/g, thus allowing a quantitative analysis with this setup. The GC-MS linear calibration appears to be more accurate, as this detector has a larger linear and dynamic range than GCxIMS. Converting GCxIMS results to increase its linear range simplifies the interpretation of its results. However, this does not increase the dynamic range.

The advantage of GCxIMS is shown during the search for degradation products in the aerosol. The GCxIMS plot shows a multitude of new substances that are not identifiable in the GC-MS data. The GCxIMS’s higher separation and detection capabilities are especially beneficial when looking for unknown substances at very low concentrations and for small molecules that fragment similarly.

The presented sampling setup enables any lab to analyze condensable substances from the gas phase. Thus, further applications can range from using high-resolution MS to identify unknown degradation products in the e-cigarette’s aerosol to trapping any volatile substances from gaseous samples. The authors encourage sharing this simple method with interested parties and have already applied it in various cases, including identifying contaminants in the outflow from a hydrogen gas generator.

## Figures and Tables

**Figure 1 molecules-28-06574-f001:**
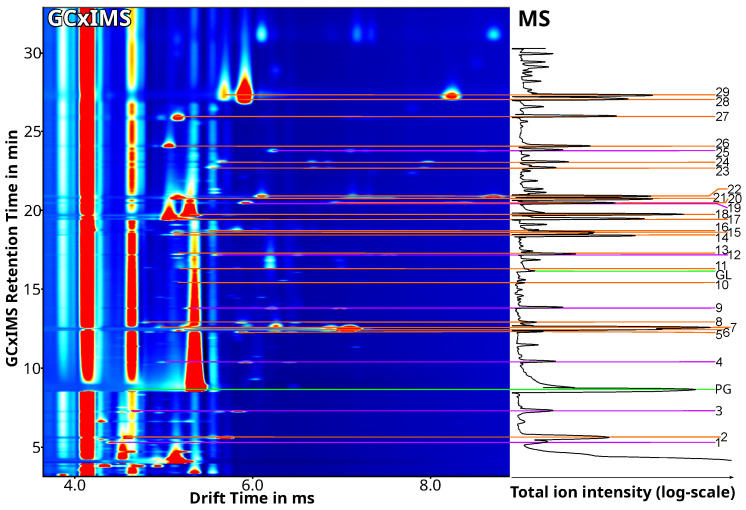
GCxIMS-plot (**left**) and the time-fitted, logarithmized GC-MS-TIC (**right**) of a single puff from a commercial, blueberry-flavored e-liquid, aerosolized using an e-cigarette at 100 °C and a flow rate of 1.5 mL/min for 5 s. The signal intensity of the GCxIMS is plotted using the colors, whereas the GC-MS signal intensity is plotted on the x-axis with a log-scale. The identified substances in both plots are connected with color-coded lines and are labeled on the far right. Matrix substances (PG, GL) are marked in green, the 2-alkanones (ISTD) are marked in purple, and the identified VOCs are marked in orange, with Table 1 listing the identified substances.

**Figure 2 molecules-28-06574-f002:**
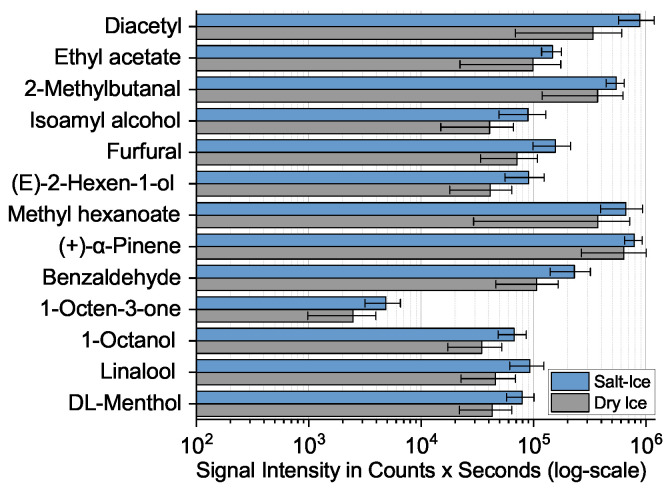
Diagram comparing the signal intensity on a logarithmic scale for 14 selected standard substances in a lab-made e-liquid collected with five repetitions either in a vial cooled in a salt–ice bath or a dry ice bath and measured by GC-MS.

**Figure 3 molecules-28-06574-f003:**
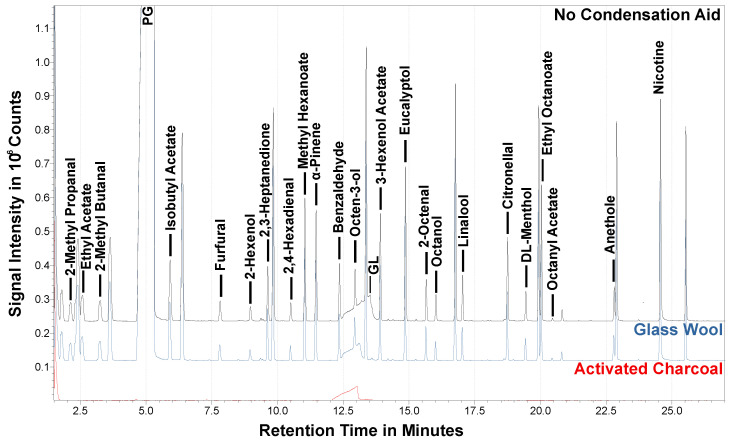
Base-shifted TICs from the GC-MS measurements of the at 100 W and 1.5 L/min aerosolized test samples collected for 5 s using activated charcoal, glass wool, or no additional condensation aid in a HS vial at about −20 °C. Selected flavor compounds are labeled, matrix substances and internal standards are unlabeled.

**Figure 4 molecules-28-06574-f004:**
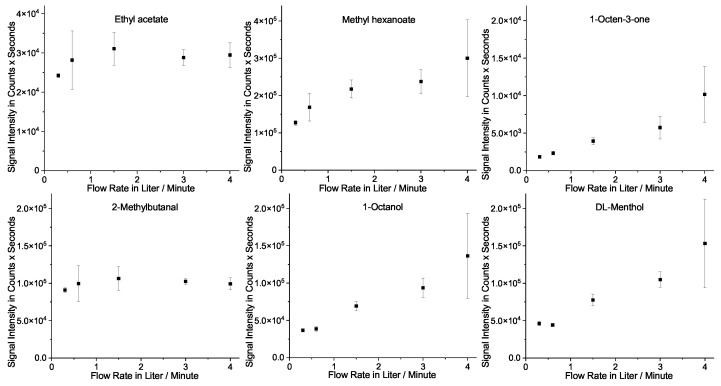
Comparison of the average signal intensities and standard deviations for six selected substances from three GC-MS measurements of the at 50 W with an e-cigarette aerosolized e-liquid (about 10 µg/g analyte in the e-liquid)) and for 5 s at −20 °C collected test samples at varying flow rates.

**Figure 5 molecules-28-06574-f005:**
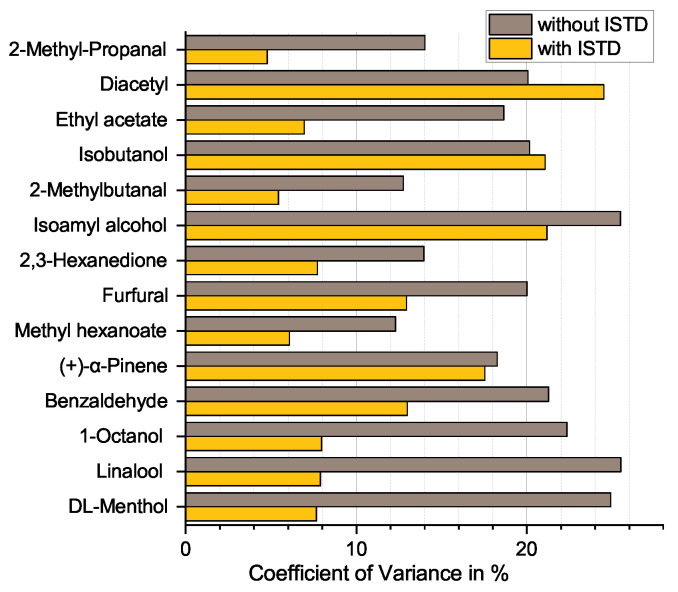
Diagram comparing the relative standard deviation (or coefficient of variance) of the signal intensity for 14 selected standard substances (about 10 µg/g) in a lab-made e-liquid for seven GC-MS measurements, each with and without the use of an internal standard (about 10 µg/g).

**Figure 6 molecules-28-06574-f006:**
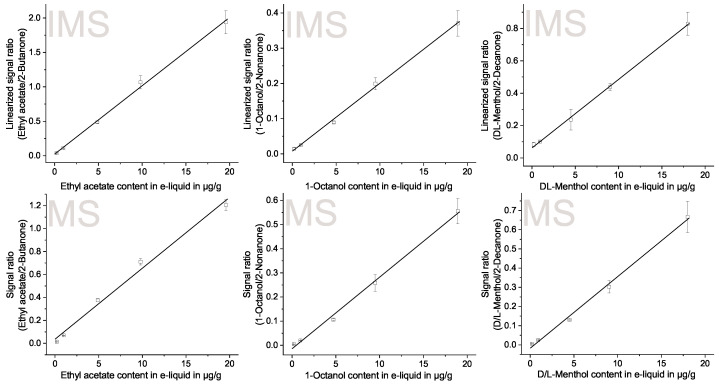
Calibration regression for ethyl acetate, 1-octanol and D/L-menthol using linearly converted GCxIMS results (**top**) and the direct GC-MS results (**bottom**) from triplicate measurements of the at 100 W and and 1.5 L/min flow aerosolized and for 5 s at −20 °C collected test samples as ratio to the appropriate ISTD (about 10 µg/g).

**Figure 7 molecules-28-06574-f007:**
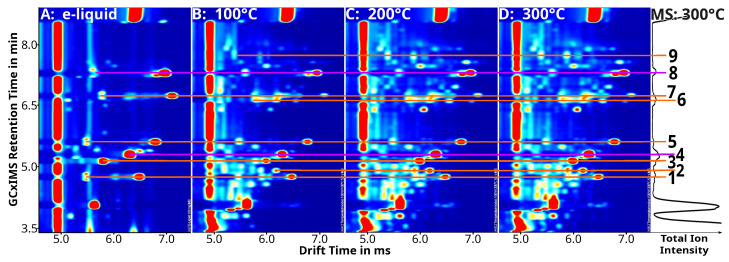
Excerpt of GCxIMS plots of a at 10 µg/g flavored lab-made e-liquid, measured directly (**A**) and as condensate (collected for 5 s at 1.5 L/min flow and −20 °C) created from the aerosol of an e-cigarette set with three different temperatures ((**B**): 100 °C, (**C**): 200 °C, (**D**): 300 °C) and a flow rate of 1.5 L/min, as well as the corresponding GC-MS TIC collected at 300 °C. The intensity of the GCxIMS is shown using colors, while the TICs intensity is plotted on the x-axis on the right using a log-scale. Signals of identified substances are connected with color-coded lines (orange: identified VOC, puple: ISTD), and are numbered on the right (1: isobutanal, 2: methacrolein*, 3: diacetyl, 4: 2-butanone, 5: ethyl acetate, 6: acetol*, 7: 2-methylbutanal, 8: 2-pentanone, 9: acetoin*; names are marked (*) when only tentatively identified using the mass spectrum).

**Figure 8 molecules-28-06574-f008:**
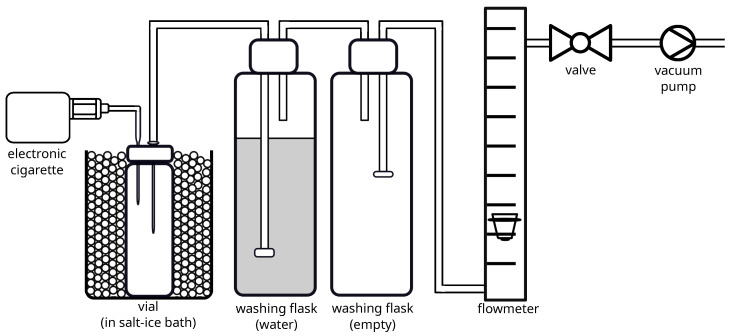
Scheme of the sampling setup, consisting of an e-cigarette connected to a cooled headspace vial (to collect the samples), two washing flasks (to protect the sampling setup from contamination), a flow meter, a valve to control the flow rate, and a vacuum pump (to provide the air flow).

**Table 1 molecules-28-06574-t001:** List of the substances, their retention times, and reduced mobilities ($K_{0}^{X}$) for monomer (${}^{M}$) and dimer (${}^{D}$) signals, which were identified in the GCxIMS and GC-MS measurement and are shown in Figure 1. Substances which where only tentatively identified using the fragmentation patterns and retention index are marked^lib^, substances added to the e-liquid and used as internal standards^istd^, substances identified and verified using a reference standard^ref^ and matrix compounds^matrix^.

Label	Name	Rt (MS)	Rt (IMS)	$K_{0}^{M}$	$K_{0}^{D}$
		/min	/min	/Vs/cm^2^	/Vs/cm^2^
1	2-Butanone^istd^	2.34	5.28	1.85	1.55
2	Ethyl acetate^ref^	2.49	5.59	1.79	1.44
3	2-Pentanone^istd^	3.52	7.28	1.75	1.40
PG	Propylene glycol^matrix^	4.67	9.13	1.77	1.53
4	2-Hexanone^istd^	6.22	10.42	1.64	1.28
5	Allyloxy-2-propanol^lib^	8.14	12.35	1.70	1.28
6	Ethyl 2-methylbutanoate^ref^	8.28	12.47	1.59	1.16
7	Ethyl 3-methylbutanoate^ref^	8.39	12.57	1.56	1.16
8	Ethyl 2-allyloxypropanoate^lib^	8.73	12.95	1.54	1.27
9	2-Heptanone^istd^	9.69	13.87	1.54	1.18
10	α-Pinene^ref^	11.32	15.60	1.60	
11	Benzaldehyde^ref^	12.22	16.55	1.54	1.33
12	2-Octanone^istd^	13.23	20.63	1.46	1.09
13	β-Pinene^ref^	14.45	18.33	1.60	
14	o-Cymene^ref^	14.46	18.58	1.60	
15	Limonene^ref^	14.61	20.42	1.59	1.16/1.12
16	Benzyl alcohol^ref^	14.69	18.87	1.49	1.07
17	Ethyl levulate^lib^	15.58	19.70	1.62	
18	Diethyl malonate^lib^	15.88	19.90	1.55	
19	2-Nonanone^istd^	16.61	20.63	1.39	1.00
20	Pentyl butanoate^lib^	16.65	20.67	1.39	0.98
21	Linalool^ref^	16.88	20.95	1.59	1.16/1.13/1.09
22	Iso-amyl isovalerate^lib^	17.05	21.10	1.35	0.94
23	Menthone^ref^	18.74	22.92	1.49	1.04
24	Menthone^ref^	18.74	23.28	1.45	1.03
25	2-Decanone^istd^	19.79	23.97	1.32	0.96
26	Methyl salicylate^lib^	20.00	24.27	1.62	
27	Linalyl acetate^lib^	21.66	26.12	1.59	
28	Ethyl 3-(2-methyl-1.3-dioxan-2-yl)propanoate^lib^	22.62	27.55	1.44	1.00
29	Ethyl 3-(2.4-dimethyl-1.3-dioxolan-2-yl)propanoate^lib^	22.82	27.77	1.39	

**Table 2 molecules-28-06574-t002:** Results for the calibration using samples collected in triplicate (aerosolized at 100 W, and 100 °C, at 1.5 mL/min flow, collected for 5 s at −20 °C) from a lab-made e-liquid with five different concentration levels, showing the root mean squared error (RMSE), the relative error (RE) and the coefficient of determination (r^2^) for the GC-MS and linearized GCxIMS results using the ratio to the appropriate ISTD (about 10 µg/g).

		GCxIMS			GC-MS		
Name	Range	RMSE	RE	r^2^	RMSE	RE	r^2^
	/µg/g	/µg/g	/%		/µg/g	/%	
2-Methylpropanal ^a^	0.18–8.1	0.53	11.45	0.973	0.51	5.43	0.992
Diacetyl ^a^	0.24–11.0	0.1	1.68	0.999	0.29	2.29	0.999
Ethyl acetate	0.21–19.6	0.36	3.6	0.997	0.71	7.08	0.990
Isobutanol	0.19–17.6	0.23	2.57	0.999	0.45	4.93	0.995
2-Methylbutanal	0.19–18.0	0.67	7.21	0.990	0.08	0.86	1.000
Isoamyl alcohol ^b^	0.20–18.2	-	-	-	0.67	7.16	0.990
Methyl isobutyl ketone ^b^	0.19–17.9	-	-	-	0.11	1.21	1.000
2,3-Hexanedione ^b^	0.22–20.6	-	-	-	0.36	3.38	0.998
Hexanal	0.20–18.7	1.39	14.33	0.960	0.11	1.14	1.000
Butyl acetate	0.22–20.0	0.58	5.62	0.994	0.14	1.36	1.000
Furfural	0.28–26.2	0.92	6.78	0.991	0.46	3.39	0.998
Ethyl 2-methyl butanoate	0.20–18.3	0.02	0.2	1.000	0.20	2.13	0.999
(E)-2-Hexenal	0.14–13.4	0.45	6.6	0.991	0.05	0.70	1.000
(E)-2-Hexen-1-ol	0.20–18.7	0.64	6.64	0.991	0.44	4.60	0.996
Ethyl pentanoate	0.21–19.5	0.5	5.01	0.995	0.24	2.44	0.999
Methyl hexanoate	0.21–19.7	0.63	6.27	0.992	0.25	2.53	0.999
(+)-α-Pinene	0.21–19.2	1.3	13.13	0.966	0.29	3.00	0.998
Benzaldehyde	0.26–23.7	0.62	5.13	0.995	0.37	3.03	0.998
1-Octen-3-one	0.20–18.9	0.67	6.88	0.991	0.20	2.02	0.999
Octanal	0.20–18.7	1.00	10.36	0.979	0.11	1.20	1.000
(E,E)-2,4-Heptadienal	0.19–17.4	0.86	9.59	0.982	0.31	3.43	0.998
D(+)-Limonene	0.20–18.9	1.54	15.71	0.952	0.47	4.86	0.995
1-Octanol	0.20–18.9	0.31	3.23	0.998	0.41	4.19	0.997
Linalool	0.21–19.1	0.63	6.45	0.992	0.38	3.87	0.997
L-Menthone ^a^	0.18–16.5	0.25	2.99	0.998	0.20	2.35	0.999
D-Menthone ^a^	0.04–3.4	0.06	3.65	0.997	0.06	3.23	0.998
D/L-Menthol	0.19–18.0	0.23	2.47	0.999	0.41	4.41	0.996
Neral ^a^	0.47–9.3	0.53	9.81	0.975	0.30	5.62	0.992
Geranial ^a^	0.12–10.9	1.51	23.51	0.868	0.35	5.52	0.992

^a^ Deviating calibrated range due to the limited dynamic range of the GCxIMS (2-methylpropanal, diacetyl) or separated enantiomers (D/L-menthone, neral, geranial), using only four different concentration levels except for D/L-menthone. ^b^ The results of the GCxIMS were unusable, as the coeluting matrix suppresses the analyte signal.

## Data Availability

Data will be made available on request.

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
