# Peer review of "Making Every Single Puff Count—Simple and Sensitive E-Cigarette Aerosol Sampling for GCxIMS and GC-MS Analysis"

_molecules, 2023, doi:10.3390/molecules28186574_

Round 1

Reviewer 1 Report

The work here presented by Alexander L. R. M. Augustini et al. is suitable for the journal and it will be of interest for the reader of Molecules. Again it represents one of the few studies on the topic totally unsponsored by companies involved in e-cig components.

As a whole I consider the study worthy to be published, however, some aspects need to be improved.

The authors should iclude more details regarding the sampling process: The authors should better explain why they chose sub-ohmic resistence and that given PG VG ratio.

In the discussion section it would be interesting if authors discuss their results also in the light of the results from these studies: PMID: 31388676 and PMID: 31629780

Author Response

Dear Reviewer 1,

We value the time and effort you volunteered to improve our work. We also appreciate those two resourceful publications which you pointed out to us. We would like to note, that the editor asked us to move the results section before the methods section, thus changing the order in which certain figures appear and changing their numbering.

Concerning the extended details on the sampling process, we expanded the descriptions on lines 129 – 133, 431 – 433, and 436 – 437, as well as the caption of Figure 8 (previously Figure 1) to explain the process more precisely.

We also added an explanation on the sub-ohmic-type of our e-cigarette in the introduction (lines 40 – 43). This includes a remark on the potential increased degradation of compounds, referencing the recommended article (no. 9, Cirillo et al., Toxicol. Sci. 2019) and added a remark about this type of e-cigarette on lines 412 – 413.

On our choice of e-liquids: the commercial, flavorless e-liquid was chosen for availability reasons. Due to legislative changes, this kind of product (containers > 10 mL, containing nicotine) is no longer available in Europe, thus we were unable to obtain other large enough quantities of a flavorless e-liquid from a single production batch, without involving the manufacturers, which we chose not to do.

The lab-made mixture of PG + GL, 1+1, was successfully used in previous publications by this work group. Thus, it was used here as well.

The studies you mentioned in articles PMID: 31388676 and PMID: 31629780 by Cirillo et al. look in depth at the toxicological effects of consuming electro-smoked cigarettes. However, this manuscript is mainly used to describe and evaluate the sampling process, presenting initial results to prove its usability.

In follow-up studies, the vaping process itself can be evaluated for various effects. We hope that with the sampling system we have developed, we can contribute to advancing these investigations. We understand further investigations are needed to fully understand the vaping process.

Thank you again for your comments. They have helped us to improve this and hopefully also future work.

Yours sincerely
A. L. R. M. Augustini
(on behalf of all authors)

Reviewer 2 Report

This manuscript deals with the development and optimisation od a sampling setup for reliably sample and analyse products in single puff in e-cigarette aerosol. This setup was devised "using readily available lab materials to allow for a simple recreation in other labs".  Moreover, the authors state that they "... encourage sharing this simple method with interested parties".

This referee can conclude that this manuscript is to be considered as a technical paper, since the aims of the authors are clearly devoted to spread their design and encourage laboratories to use for analysis of e-cigarettes.

In this view, I think that the chemical characterisation of reaction and decomposition products in the e-cigarette aerosol formation is useless in this approach and should be removed.  In this way, the authors will have the space and motivation to introduce and discuss the use of an insulated or heated transfer line into the vial, which is well-known to them to give the opportunity of "reduce the condensation outside the vial, thus improving the sampling conditions." (see lines 294-297).

In conclusion, this manuscript has to be thoroughly revised, focussing results and discussion to the optimisation of their experimental setup, which I consider of potential interests for laboratories.

Author Response

Dear Reviewer 2,

We appreciate the time and effort you volunteered to improve our work. We would like to note, that the editor asked us to move the results section before the methods section, thus changing the order in which certain figures appear and changing their numbering.  

You pinpoint correctly, that this manuscript presents a new sampling setup and process which is then evaluated for its validity and applicability. We strongly support publicly sharing research results and methods. For this reason, submitted this manuscript to a widespread open-access-only journal.

The observed decomposition products are described and evaluated to show the capabilities of this simple but powerful setup. It is not meant as a comprehensive study of the degradation, thus the presented results are compared to published results and the wording used to describe them is deliberately cautious. Regardless, we found these results to be very graphic and easily understandable, thus a good medium to present the possibilities of this method. To address your concerns, we reworked the entire paragraph to be more cautious about our tentative results. See lines 383 – 399 for the improved segment.

The proposed use of a heated transfer line is just a suggestion to improve the sample collection for further work if a higher sensitivity is needed or more volatile compounds should be trapped. Such a setup would negate the goal of a simple setup as such a heated transfer line is a custom-build item, exceeding the capabilities of most labs.

Additionally, the collection setup and analysis were matched with each other. Changing the trapping capacity would also require the adaptation of the analytical systems for higher volatile compounds. This would require extending previous projects on correlation, identification, and quantitation to include these compounds. We are aware of the possible interest in these capabilities, however, expanding the project to cover these compounds exceeds its current scope. Thus, at this moment we need to decline your encouragement on the enhancement of our setup and hope to be able to include this in future research.

Thank you again for your detailed comments. They have helped us to revisit the manuscript and refine certain aspects. We hope you understand our reasoning and support publishing the results as presented.

Yours sincerely
A. L. R. M. Augustini
(on behalf of all authors)

Reviewer 3 Report

In this paper, a complete experimental device was set up to extract samples of e-cigarette aerosol, and it was detected by GC-MS and GCxIMS. I have some comments which the authors may find beneficial.

1.         In this paper, the device for extracting e-cigarette aerosol is described emphatically, but the principle of measuring methods such as GCxIMS is not introduced in detail, which needs to be modified and adjusted appropriately.

2.         Regarding the abbreviation of gas chromatography coupled ion mobility spectroscopy (GCxIMS), can x be replaced with × Or -, easily misleading.

3.         In Figure 1, what is the function of the first vial (in salt-ice bath)? And during the gas washing process, the product of the electronic cigarette contains water vapor, does the process has an impact on its content? It is better to number these vials and explain the function of each vial.

4.         On page 7, about the “The drift and retention times of the substances were determined to be very stable (±4 µs and ±1 s, respectively) over a period of 60 days”, what is the significance of 60 days? Is the sample that has been stored for 60 days used for GCxIMS?

5.         On page 9, the glass wool adding to the bottle may adsorb the e-cigarette aerosol and lead to the loss of some particles. After all, e-cigarettes produce not pure steam, but smoke that may contain certain particles.

6.         Some statement about the detection of the smokes should be revised with proper citation: DOI: 10.1016/j.chemosphere.2019.125184; 10.1364/OE.27.00A790; 10.1016/j.ijleo.2023.170867; 10.1016/j.ijleo.2021.166999.

7.         On page 13, in the process of analyzing e-cigarette samples by GC-MS and GCxIMS, the temperature is set at 100℃. However, the temperature is not 100℃ when the e-cigarette is used for smoking normally. Thus the substances in the sample should be different as their original properties.

Author Response

Dear Reviewer 3

We appreciate the time and effort you volunteered to improve our work. We would like to note, that the editor asked us to move the results section before the methods section, thus changing the order in which certain figures appear and changing their numbering. We like to address all your remarks in-depth and separately in the following passages:

"1. In this paper, the device for extracting e-cigarette aerosol is described emphatically, but the principle of measuring methods such as GCxIMS is not introduced in detail, which needs to be modified and adjusted appropriately."

Thank you for your concern about the comprehensiveness of this work. You are correct, that GC-coupled ion mobility spectrometry is not as well-known as GC-MS, however, it is an established technology with early publications from the 1970s. To ease into the topic, we have added a paragraph on the common use at airports, which is its most recognizable use (lines 102 – 103).

The working principle, the field of application, and current developments are summarized in two paragraphs on lines 90 – 109, in the introduction. Of course, it is not possible to include a complete explanation of this technology, without exceeding the appropriate length of the introduction for research that used this method only as a measuring tool.
When you are interested in a more detailed summary, we recommend the two-part review from Cumeras at al. (doi.org/10.1039/c4an01100g and doi.org/10.1039/c4an01101e).

"2. Regarding the abbreviation of gas chromatography coupled ion mobility spectroscopy (GCxIMS), can x be replaced with × Or -, easily misleading."

This is a much-discussed topic in this field. The most common choices are “-” and “x”. In our experience, the “×” is rarely seen, probably because special characters are difficult to use in keyword-related searches. We have chosen “x”, as it describes best the orthogonality of the coupled separation techniques (GC and IMS) and refers to the three-dimensional results that are created. The “-” is commonly used for technologies combining one-dimensional separation with one-dimensional detection, resulting in two-dimensional data.

"3. In Figure 1, what is the function of the first vial (in salt-ice bath)? And during the gas washing process, the product of the electronic cigarette contains water vapor, does the process has an impact on its content? It is better to number these vials and explain the function of each vial."

Figure 8 (previously Figure 1) shows only a single vial, that is used to collect the sample. The washing bottles are used to protect the flow line and vacuum pump from being contaminated by the sample. We have clarified this in the caption below the figure.

"4. On page 7, about the “The drift and retention times of the substances were determined to be very stable (±4 µs and ±1 s, respectively) over a period of 60 days”, what is the significance of 60 days? Is the sample that has been stored for 60 days used for GCxIMS?"

The stability has been observed for a period of 60 days. This refers to the unchanged parameters needed for the identification of compounds using the GCxIMS. As it is comparable to a calibration, this needs to be regularly verified using control standards. Our mentioning of this fact is meant as a data point for other users to compare their stability to and as recognition of a parameter that needs to be verified during use. To clarify this point, we added a short paragraph in the results section (lines 165 – 167).

"5. On page 9, the glass wool adding to the bottle may adsorb the e-cigarette aerosol and lead to the loss of some particles. After all, e-cigarettes produce not pure steam, but smoke that may contain certain particles."

Thank you for your remark on particles in tobaccoless e-cigarettes. We are aware of the formation and analysis of these particles as described in multiple publications focusing on this topic. You are correct, that the particles’ surface will interact with the volatile analytes, thus influencing the composition of the collected sample. For this reason, we match the matrix of the e-liquids and vaping parameters for our calibrations as closely as possible to include these effects as well. Nonetheless, we noted the influence of particles in the subsection “Condensation Aids” (lines 209 – 210).

"6. Some statement about the detection of the smokes should be revised with proper citation: DOI: 10.1016/j.chemosphere.2019.125184; 10.1364/OE.27.00A790; 10.1016/j.ijleo.2023.170867; 10.1016/j.ijleo.2021.166999 ."

Thank you for pointing out these very interesting publications on using laser-induced breakdown spectroscopy for the detection of elements and small inorganic compounds in smoke. This can be a promising technique for the detection of metal contaminants in vaped e-liquids or the decomposition of metal-based heating coils of e-cigarettes. However, our current manuscript focuses on the detection of volatile organic compounds using gas chromatography. Thus, we are uncertain, where we can include these citations in our current manuscript as we haven’t touched on this field of inquiry yet. So, thank you again for these intriguing publications, which we will keep in mind for our future research.

"7. On page 13, in the process of analyzing e-cigarette samples by GC-MS and GCxIMS, the temperature is set at 100℃. However, the temperature is not 100℃ when the e-cigarette is used for smoking normally. Thus the substances in the sample should be different as their original properties."

We are aware of the various possible temperature effects when a tobaccoless e-cigarette is used. The mentioned temperatures are part of the settings for the e-cigarette and we assume this to be a goal setting for the coil itself, as the evaporation of propylene glycol, water, and glycerine cool the wicking material and coil. This has been evaluated in detail by Geiss et al. (doi.org/10.1016/j.ijheh.2016.01.004).

We have looked into the effect of the set temperature in subsection 2.7: „Reactivity and Degradation“. The difference between the e-liquid and the aerosol produced at the different set temperatures is shown there. Further investigations into the effects of the temperature, especially in regard to the findings of Geiss et al. concerning the differences between the successive puffs, can be the focus of future research.

Thank you again for your detailed comments. They have helped us to enhance the manuscript, which we hope to now match your requirements.

Yours sincerely
A. L. R. M. Augustini
(on behalf of all authors)

Round 2

Reviewer 2 Report

All my recommendations seem to be correctly addressed.

Tha manuscript can be published. 

Reviewer 3 Report

The authors have addressed all the points.